# Association of race and health insurance in treatment disparities of colon cancer: A retrospective analysis utilizing a national population database in the United States

Scarlett Hao [1], Rebecca A. Snyder [2,3], William Irish [1,3], Alexander A. Parikh [2]*

1 Department of Surgery, Brody School of Medicine at East Carolina University, Greenville, North Carolina, United States of America, 2 Division of Surgical Oncology, Department of Surgery, Brody School of Medicine at East Carolina University, Greenville, North Carolina, United States of America, 3 Department of Public Health, East Carolina University, Greenville, North Carolina, United States of America

* parikha19@ecu.edu

## Abstract

**Data Availability Statement:** The data underlying the results presented in the study are available

### Background

Both health insurance status and race independently impact colon cancer (CC) care delivery and outcomes. The relative importance of these factors in explaining racial and insurance disparities is less clear, however. This study aimed to determine the association and interaction of race and insurance with CC treatment disparities.

### Study setting

Retrospective cohort review of a prospective hospital-based database.

### Methods and findings

In this cross-sectional study, patients diagnosed with stage I to III CC in the United States were identified from the National Cancer Database (NCDB; 2006 to 2016). Multivariable regression with generalized estimating equations (GEEs) were performed to evaluate the association of insurance and race/ethnicity with odds of receipt of surgery (stage I to III) and adjuvant chemotherapy (stage III), with an additional 2-way interaction term to evaluate for effect modification. Confounders included sex, age, median income, rurality, comorbidity, and nodes and margin status for the model for chemotherapy. Of 353,998 patients included, 73.8% (n = 261,349) were non-Hispanic White (NHW) and 11.7% (n = 41,511) were non-Hispanic Black (NHB). NHB patients were less likely to undergo resection [odds ratio (OR) 0.66, 95% confidence interval [CI] 0.61 to 0.72, p < 0.001] or to receive adjuvant chemotherapy [OR 0.83, 95% CI 0.78 to 0.87, p < 0.001] compared to NHW patients. NHB patients with private or Medicare insurance were less likely to undergo resection [OR 0.76, 95% CI 0.63 to 0.91, p = 0.004 (private insurance); OR 0.59, 95% CI 0.53 to 0.66, p < 0.001 (Medicare)] and to receive adjuvant chemotherapy [0.77, 95% CI 0.68 to 0.87, p < 0.001 (private insurance); OR 0.86, 95% CI 0.80 to 0.91, p < 0.001 (Medicare)] compared to similarly

from the National Cancer Database ([https://www.facs.org/quality-programs/cancer/ncdb](https://www.facs.org/quality-programs/cancer/ncdb)).

**Funding:** The author(s) received no specific funding for this work.

**Competing interests:** I have read the journal's policy and the authors of this manuscript have the following competing interests: Authors RS and AP are spouses.

**Abbreviations:** ACA, Affordable Care Act; AJCC, American Joint Commission on Cancer; CC, colon cancer; CI, confidence interval; CoC, Commission on Cancer; GEE, generalized estimating equation; MIS, minimally invasive surgery; NCDB, National Cancer Database; NHB, non-Hispanic Black; NHW, non-Hispanic White; OR, odds ratio; SDOH, social determinants of health; SEER, Surveillance Epidemiology and End Results; STROBE, Strengthening the Reporting of Observational Studies in Epidemiology.

insured NHW patients. Although Hispanic patients with private and Medicare insurance were also less likely to undergo surgical resection, this was not the case with adjuvant chemotherapy. This study is mainly limited by the retrospective nature and by the variables provided in the dataset; granular details such as continuity or disruption of insurance coverage or specific chemotherapy agents or dosing cannot be assessed within NCDB.

## Conclusions

This study suggests that racial disparities in receipt of treatment for CC persist even among patients with similar health insurance coverage and that different disparities exist for different racial/ethnic groups. Changes in health policy must therefore recognize that provision of insurance alone may not eliminate cancer treatment racial disparities.

## Author summary

### Why was this study done?

- Patients of Black and Hispanic race and ethnicity have a higher incidence of colon cancer (CC), are diagnosed with more advanced disease, and have poorer survival than White patients.

- Patients with Medicaid insurance and those without insurance also present with more advanced disease and have poorer outcomes.

- The role of insurance status in explaining these racial disparities is not well understood.

### What did the researchers do and find?

- We identified patients diagnosed with stage I to III CC within the National Cancer Database (NCDB) from 2006 to 2016.

- We investigated factors associated with receiving surgical removal of the cancer as well as chemotherapy after resection.

- We found that Black patients were less likely to undergo surgical removal and receive chemotherapy, and Hispanic patients were less likely to undergo surgical removal controlling for insurance type.

- We also found that patients with Medicaid and those without insurance also were less likely to undergo surgical removal and receive chemotherapy.

- We also found that even in patients with private and Medicare insurance, those that were Black or Hispanic were less likely to undergo surgical removal and that those that were Black also were less likely to receive chemotherapy after removal.

**What do these findings mean?**

- Results from this study suggest that even with private and Medicare insurance, certain underrepresented and underprivileged minorities such as Blacks and Hispanics are still less likely to receive standard of care for CC.

- Simply providing these patients with health insurance alone may not be enough to reduce these disparities.

- Different minorities, such as Blacks and Hispanics, have different disparities in regard to CC treatment.

- Additional research needs to be performed to identify factors that are preventing Blacks and Hispanics from receiving the standard of care for CC outside of health insurance.

## Introduction

Over 100,000 new cases of colon cancer (CC) will be diagnosed in 2021, with the highest incidence among non-Hispanic Black (NHB) patients [1]. Overall, patients of NHB and Hispanic race/ethnicity have a higher incidence of CC, are diagnosed with more advanced disease, and experience worse overall survival compared to patients of non-Hispanic White (NHW) race [1]. It has been estimated that the increase in CC mortality among Black patients may be secondary to more advanced or later stage disease at presentation [2]. This is likely also strongly influenced by social determinants of health (SDOH), which can include but are not limited to education level, employment, income level or poverty, and housing or homelessness [2]. Interventions focused on eliminating racial disparities in screening rates by overcoming some of these barriers have shown improvement in, and in some cases, even elimination of regional racial disparities in cancer outcomes [3].

Passage of the Affordable Care Act (ACA) in 2010 aimed to reduce disparities in insurance coverage with the goal of improving overall access to healthcare, including preventative care [4]. Following implementation of the ACA, health insurance coverage, screening rates, and the frequency of physician visits increased for patients of NHB and Hispanic race/ethnicity [5,6]. However, despite these improvements, minority patients still face delays in cancer treatment and are less likely to receive appropriate therapy [7–9]. It has been proposed that disparities in care may be related to environmental, lifestyle, cultural, socioeconomic, behavioral, and biologic factors as well as access to quality healthcare [10]. Ultimately, however, the intersection between racial disparities in treatment and insurance status remains poorly understood.

The primary aim of this study was to evaluate this intersection between race/ethnicity and insurance, specifically to determine whether racial/ethnic disparities in the receipt of CC treatment potentially differ among patients with the same insurance coverage.

## Methods

### Data source

The National Cancer Database (NCDB), sponsored by the American College of Surgeons and American Cancer Society, gathers data from more than 1,500 Commission on Cancer (CoC)-accredited facilities in the US. CoC cancer registrars are trained and certified to code data

according to rigorously established protocols. The NCDB includes data on more than 70% of newly diagnosed cancer cases nationwide and is felt to be representative of national practice patterns in cancer care [11]. This study was reviewed by the Institutional Review Board of the Brody School of Medicine at East Carolina University and determined to be exempt. Results are reported per the Strengthening the Reporting of Observational Studies in Epidemiology (STROBE) reporting guidelines [12].

## Study population

Patients aged 18 years or older with a new diagnosis of stage I, II, or III adenocarcinoma of the colon, as defined by the American Joint Commission on Cancer (AJCC), between 2006 and 2016 were identified from the NCDB. Stage was defined according to the sixth and seventh edition of the *AJCC Cancer Staging ManuaI* [13,14]. Patients with a prior cancer diagnosis were excluded. Patients were then divided into cohorts by race/ethnicity for comparison. Patient race and ethnicity were determined from predefined NCDB data based on assignment by a CoC registrar according to fixed categories, specifically NHW, NHB, Hispanic, and Other.

## Variables and outcomes

Clinical and demographic variables were selected a priori from the available data provided in the NCDB participant user file. These included age, race, ethnicity, sex, primary payor, median household income, educational attainment (number of adults in the patient's ZIP code who did not graduate from high school), rural/urban residence, distance traveled for care, and Charlson/Deyo comorbidity index. Cancer-specific variables included primary tumor location, histologic grade, and analytic stage based on the AJCC classification sixth and seventh edition. Primary tumor location was categorized as left (splenic flexure, descending, or sigmoid), right (cecum, ascending, hepatic flexure, or transverse), or overlapping/not otherwise specified. The design and analysis plan for the study is shown in the Supporting information (S1 Table). The primary outcomes of interest were (1) receipt of surgical resection; and (2) receipt of adjuvant chemotherapy in the subgroup of eligible patients with resected stage III CC, stratified by race/ethnicity and insurance.

## Statistical analysis

Continuous variables are described by the number of nonmissing observations, mean, standard deviation, median, and 25th and 75th percentiles. Categorical variables are described overall and by cohort with the number of patients and percentage for each category. Missing data were considered as a separate category.

Outcomes of receipt of surgery and receipt of chemotherapy were stratified by race/ethnicity and insurance and presented as unadjusted percentages. Comparisons were made using chi-squared, 1-way ANOVA, and Kruskal–Wallis tests as appropriate. To adjust for confounding and estimate the association of outcome to covariates, data were fit using multivariable binary logistic regression models. Generalized estimating equations (GEEs) approach was used to accommodate facility clustering assuming an exchangeable working correlation structure. Two GEE models were fit to the data: a main effects model with additive terms for race and insurance status adjusted for additional covariates and a joint effects model with a 2-way interaction term for race and insurance also adjusted for additional covariates. These included age, race, sex, insurance status, income level, education, rurality, comorbidity, distance traveled for care, and tumor stage. For the analysis of receipt of adjuvant chemotherapy outcome, the GEE models also included surgical margins status and the number of lymph nodes

resected. Parameter estimates were tested using the Z score. The standard errors, confidence intervals (CIs), Z scores, and *p*-values are based on empirical standard error estimates. The joint effects model was used to evaluate the effect of race on outcome within levels of insurance status. Adjusted odds ratios (ORs) and 95% CIs are provided as measures of strength of association and precision, respectively. The joint effect of race and insurance status on outcomes was tested using the generalized score chi-squared on 12 degrees of freedom. A 2-sided *p*-value <0.05 was considered statistically significant. Missing/unknown data were excluded in the multivariable analyses. Analyses were performed with SAS statistical software (version 9.4, SAS Institute, Cary, North Carolina, US).

## Results

### Demographics

Of the 908,503 patients with CC identified in the 2006 to 2016 NCDB participant user file, 353,998 patients met inclusion criteria (**Fig 1**). The subgroup of patients with stage III disease assessed for receipt of adjuvant chemotherapy totaled 129,341 patients. Demographic data by racial cohorts are demonstrated in **Table 1.** There were some small differences in regard to mean age at diagnosis and sex across groups. Clinical characteristics were also somewhat different among the cohorts, including Charlson/Deyo comorbidity index as well as primary tumor location (right-sided tumor; 61.1% NHW versus 59.8% NHB, *p* < 0.001). AJCC stage distribution also varied among the cohorts, with approximately 28.1% stage I, 35.2% stage II, and 36.2% stage III among NHW and 27.5% stage I, 32.7% stage II, and 39.8% stage III among NHB (*p* < 0.001).

Socioeconomic differences were also observed between cohorts. Compared to NHW patients, more NHB patients were uninsured (6.2% versus 2.1%, *p* < 0.001) or Medicaid insured (9.5% versus 3.0%, *p* < 0.001). Similarly, more Hispanic patients were uninsured (8.9% versus 2.1%, *p* < 0.001) or Medicaid insured (12.3% versus 3.0%, *p* < 0.001) compared to NHW patients. More NHB patients compared to NHW patients resided in a region with lower median income (45.1% versus 15.1% with median income <US$40,227, *p* < 0.001) and lower education level (40.9% versus 16.3% residing in a ZIP code in which ≥17.6% did not graduate from high school, *p* < 0.001). Hispanic patients were also more likely to reside in metropolitan areas compared to NHW patients (93.7% versus 80.6%, *p* < 0.001).

### Receipt of therapy

Among the entire cohort, 347,206 patients (98.08%) underwent surgery, with a mean time to treatment of 16.3 days (SD 28.4) (**Table 2**). Patients across all racial/ethnic cohorts had similar rates of surgery; however, NHB patients had slightly longer time to surgery compared to NHW patients (18.1 versus 15.9 days, *p* < 0.001). Of the subgroup of patients with stage III CC who underwent definitive resection, only 68.4% (*N* = 88,489) received adjuvant chemotherapy, at a mean of 52 days from resection to start of treatment. When evaluated by race/ethnic group, 67.6% of patients of NHW race received adjuvant chemotherapy compared to 70.9% of patients of NHB race (*p* < 0.001). NHB patients had a slightly longer time from surgery to the start of chemotherapy compared to NHW patients (50.1 versus 56.0 days, *p* < 0.001).

On unadjusted univariate regression analyses, race/ethnic groups were less likely to receive surgery compared to patients of NHW race but were more likely to receive adjuvant chemotherapy compared to patients of NHW race (**Table 3**). All other insurance categories were associated with lower likelihood of receipt of resection or chemotherapy compared to the private insurance category.

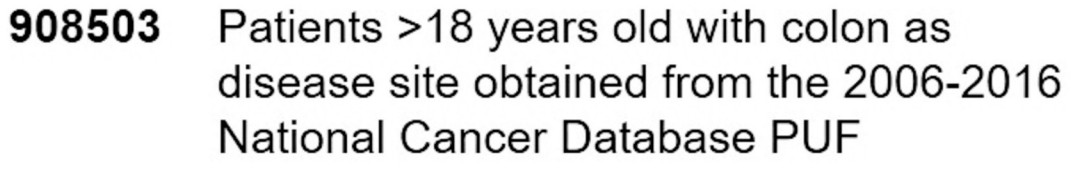

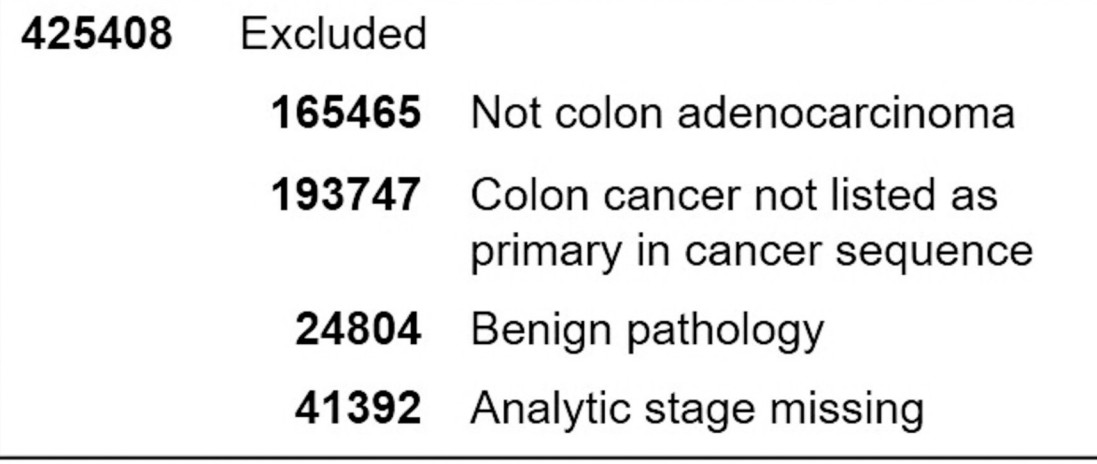

Fig 1. **Flow diagram of cohort selection. PUF, participant user file.**

**Table 1. Cohort demographics by race/ethnicity, 2006 to 2016.**

| | Racial/ethnic group | | | | | p-Value |
|---|---|---|---|---|---|---|
| | NHW n = 261,349 | NHB n = 41,511 | Hispanic n = 18,835 | Other n = 32,303 | Overall n = 353,998 | |
| Age at diagnosis (years) listed as mean (SD) | 69.58 (12.53) | 64.46 (12.04) | 64.95 (12.58) | 67.82 (12.68) | 68.57 (12.63) | p < 0.001 |
| **Insurance** | | | | | | |
| Uninsured | 5,503 (2.1%) | 2,588 (6.2%) | 1,682 (8.9%) | 1,100 (3.4%) | 10,873 (3.1%) | p < 0.001 |
| Medicaid | 7,822 (3.0%) | 3,953 (9.5%) | 2,322 (12.3%) | 2,082 (6.5%) | 16,179 (4.6%) | |
| Medicare | 158,536 (60.7%) | 19,410 (46.8%) | 8,028 (42.6%) | 16,600 (51.4%) | 202,574 (57.2%) | |
| Private | 87,311 (33.4%) | 15,107 (36.4%) | 6,681 (35.5%) | 12,189 (37.7%) | 121,288 (34.3%) | |
| Other government | 2,177 (0.8%) | 453 (1.1%) | 122 (0.7%) | 332 (1.0%) | 3,084 (0.9%) | |
| **Sex** | | | | | | p < 0.001 |
| Female | 135,924 (52.0%) | 23,212 (55.9%) | 9,251 (49.1%) | 16,957 (54.5%) | 185,344 (52.4%) | |
| Male | 125,425 (48.0%) | 18,299 (44.1%) | 9,584 (50.9%) | 15,346 (47.5%) | 168,654 (47.6%) | |
| **Median household income** | | | | | | p < 0.001 |
| Less than US$40,227 | 39,330 (15.1%) | 18,699 (45.1%) | 5,110 (27.1%) | 5,235 (16.2%) | 68,374 (19.3%) | |
| US$40,228 to US$50,353 | 59,548 (22.8%) | 8,359 (20.1%) | 4,361 (23.2%) | 6,542 (20.3%) | 78,810 (22.3%) | |
| US$50,354 to US$63,332 | 63,282 (24.2%) | 6,326 (15.2%) | 4,412 (23.4%) | 7,418 (23.0%) | 81,438 (23.0%) | |
| US$63,333+ | 95,441 (36.5%) | 7,445 (17.9%) | 4,740 (25.2%) | 12,747 (39.5%) | 120,373 (34.0%) | |
| Not available | 3,748 (1.4%) | 682 (1.6%) | 212 (1.1%) | 361 (1.1%) | 5,003 (1.4%) | |
| **% did not graduate from HS** | | | | | | p < 0.001 |
| Less than 6.3% | 68,457 (26.2%) | 3,376 (8.1%) | 1,727 (9.2%) | 819 (25.4%) | 81,753 (23.1%) | |
| 6.3% to 10.8% | 78,959 (30.2%) | 7,459 (18.0%) | 2,935 (15.6%) | 9,103 (28.2%) | 98,456 (27.8%) | |
| 10.9% to 17.5% | 68,057 (26.0%) | 13,081 (31.5%) | 3,896 (20.7%) | 8,091 (25.1%) | 93,125 (26.3%) | |
| 17.6% or more | 42,683 (16.3%) | 16,982 (40.9%) | 10,082 (53.5%) | 6,607 (20.5%) | 76,354 (421.6%) | |
| Not available | 3,193 (1.2%) | 613 (1.5%) | 195 (1.0%) | 309 (1.0%) | 4,310 (1.2%) | |
| **Rurality** | | | | | | p < 0.001 |
| Metro | 210,691 (80.6%) | 36,998 (89.1%) | 17,654 (93.7%) | 27,341 (84.6%) | 292,684 (82.7%) | |
| Urban | 38,396 (14.7%) | 3,369 (8.1%) | 784 (4.2%) | 3,610 (11.2%) | 46,159 (13.0%) | |
| Rural | 5,445 (2.1%) | 454 (1.1%) | 48 (0.3%) | 679 (2.1%) | 6,626 (1.9%) | |
| Not available | 6,817 (2.6%) | 690 (1.7%) | 349 (1.9%) | 673 (2.1%) | 8,529 (2.4%) | |
| **Distance traveled for care** | | | | | | p < 0.001 |
| Mean (SD) | 23.25 (93.92) | 14.68 (60.03) | 16.87 (75.84) | 21.99 (110.68) | 21.79 (91.47) | |
| 25th to 75th | 3.90 to 18.90 | 3.00 to 12.20 | 3.10 to 11.90 | 3.50 to 15.40 | 3.70 to 17.40 | |
| Median | 8.30 | 6.20 | 6.20 | 7.20 | 7.80 | |
| **Charlson/Deyo comorbidity index** | | | | | | p < 0.001 |
| 0 | 175,197 (67.0%) | 27,342 (65.9%) | 12,857 (68.3%) | 22,504 (69.7%) | 237,900 (67.2%) | |
| 1 | 58,370 (22.3%) | 9,819 (23.7%) | 4,357 (23.1%) | 6,870 (21.3%) | 79,416 (22.4%) | |
| 2 | 18,498 (7.1%) | 2,802 (6.8%) | 1,042 (5.5%) | 2,010 (6.2%) | 24,352 (6.9%) | |
| 3 or more | 9,284 (3.6%) | 1,548 (3.7%) | 579 (3.1%) | 919 (2.8%) | 12,330 (3.5%) | |
| **Facility type** | | | | | | p < 0.001 |
| Community | 35,265 (13.5%) | 3,899 (9.4%) | 1,926 (10.2%) | 3,879 (12.0%) | 44,969 (12.7%) | |
| Comprehensive | 127,134 (48.7%) | 15,706 (37.8%) | 7,960 (42.3%) | 13,881 (43.0%) | 164,681 (46.5%) | |
| Academic | 61,016 (23.4%) | 15,318 (36.9%) | 6,078 (32.3%) | 9,671 (29.9%) | 92,083 (26.0%) | |
| Integrated network | 37,934 (14.5%) | 6,588 (15.9%) | 2,871 (15.2%) | 4,872 (15.1%) | 52,265 (14.8%) | |
| **Primary site** | | | | | | p < 0.001 |
| Right | 159,760 (61.1%) | 24,822 (59.8%) | 10,354 (55.0%) | 17,689 (54.8%) | 212,625 (60.1%) | |
| Left | 93,789 (35.9%) | 15,198 (36.6%) | 7,892 (41.9%) | 13,532 (41.9%) | 130,411 (36.8%) | |
| Overlapping/NOS | 7,800 (3.0%) | 1,491 (3.6%) | 589 (3.1%) | 1,082 (3.4%) | 10,962 (3.1%) | |
| **Grade** | | | | | | p < 0.001 |

(*Continued*)

**Table 1.** (Continued)

| | Racial/ethnic group | | | | | p-Value |
|---|---|---|---|---|---|---|
| | NHW *n* = 261,349 | NHB *n* = 41,511 | Hispanic *n* = 18,835 | Other *n* = 32,303 | Overall *n* = 353,998 | |
| 1 | 27,892 (10.7%) | 4,694 (11.3%) | 2,037 (10.8%) | 3,428 (10.6%) | 38,051 (10.7%) | |
| 2 | 171,698 (65.7%) | 28,747 (69.2%) | 12,583 (66.8%) | 21,720 (67.2%) | 234,748 (66.3%) | |
| 3 | 42,506 (16.3%) | 5,009 (12.1%) | 2,822 (15.0%) | 4,953 (15.3%) | 55,290 (15.6%) | |
| 4 | 7,074 (2.7%) | 662 (1.6%) | 394 (2.1%) | 568 (1.8%) | 8,698 (2.5%) | |
| Not available | 12,179 (4.7%) | 2,399 (5.8%) | 999 (5.3%) | 1,634 (5.1%) | 17,211 (4.9%) | |
| **AJCC stage** | | | | | | *p* < 0.001 |
| I | 73,420 (28.1%) | 11,410 (27.5%) | 4,675 (24.8%) | 8,861 (27.5%) | 98,366 (27.8%) | |
| II | 93,215 (35.7%) | 13,567 (32.7%) | 6,514 (34.6%) | 11,221 (34.7%) | 124,517 (35.2%) | |
| III | 94,714 (36.2%) | 16,534 (39.8%) | 7,646 (40.6%) | 12,221 (37.8%) | 131,115 (37.0%) | |

AJCC, American Joint Committee on Cancer; HS, high school; NHB, non-Hispanic Black; NHW, non-Hispanic White; NOS, not otherwise specified; SD, standard deviation.

## Multivariable logistic regression: Main effects

NHB and Hispanic race/ethnicity were independently associated with decreased odds of undergoing surgical resection compared to NHW race [OR 0.66, 95% CI 0.61 to 0.72 (NHB); OR 0.76, 95% CI 0.67 to 0.85 (Hispanic)] (**Fig 2**). Other factors independently associated with decreased odds of resection included Medicaid insurance (OR 0.54, 95% CI 0.47 to 0.62) and higher Charlson/Deyo comorbidity index (OR 0.73, 95% CI 0.65 to 0.81, score of 3 or more versus 0). Compared to private insurance, patients with Medicare insurance had higher odds of undergoing surgical resection (OR 1.19, 95% CI 1.11 to 1.28) (**Table 4**).

**Table 2. Receipt of treatment by insurance and race/ethnicity.**

| Cohort | | Surgery | | | | Chemotherapy | | | |
|---|---|---|---|---|---|---|---|---|---|
| | | No *n* (%) | Yes *n* (%) | Treatment started, days from Dx [mean (SD)] | *p*-Value | No *n* (%) | Yes *n* (%) | Treatment started, days from surgery [mean (SD)] | *p*-Value |
| OVERALL | | 6,792 (1.9%) | 347,206 (98.1%) | 16.3 (28.4) | *p* < 0.001 | 40,852 (31.6%) | 88,489 (68.4%) | 51.2 (34.6) | *p* < 0.001 |
| Primary payor | Uninsured | 258 (2.4%) | 10,615 (97.6%) | 13.9 (31.6) | *p* < 0.001 | 1,090 (22.6%) | 3,727 (77.4%) | 59.0 (41.2) | *p* < 0.001 |
| | Medicaid | 392 (2.4%) | 15,787 (97.6%) | 18.2 (45.8) | *p* < 0.001 | 1,664 (24.3%) | 5,180 (75.7%) | 48.1 (31.6) | *p* < 0.001 |
| | Medicare | 4,594 (2.3%) | 197,980 (97.7%) | 16.4 (27.4) | *p* < 0.001 | 30,161 (43.5%) | 39,127 (56.5%) | 58.0 (37.7) | *p* < 0.001 |
| | Private | 1,463 (1.2%) | 119,825 (98.8%) | 15.9 (26.6) | *p* < 0.001 | 7,636 (16.2%) | 39,585 (83.8%) | 52.8 (36.0) | *p* < 0.001 |
| | Other government | 85 (2.8%) | 2,999 (97.2%) | 17.1 (32.5) | *p* < 0.001 | 301 (25.7%) | 870 (74.3%) | 51.5 (28.2) | *p* < 0.001 |
| Race/ ethnicity | NHW | 4,697 (1.8%) | 256,652 (98.2%) | 15.9 (26.7) | *p* < 0.001 | 30,267 (32.4%) | 63,250 (67.6%) | 50.1 (33.5) | *p* < 0.001 |
| | NHB | 1,030 (2.5%) | 40,481 (97.5%) | 18.1 (37.5) | *p* < 0.001 | 4,735 (29.1%) | 11,543 (70.9%) | 56.0 (38.6) | *p* < 0.001 |
| | Hispanic | 424 (2.3%) | 18,411 (97.7%) | 18.0 (31.3) | *p* < 0.001 | 2,146 (28.5%) | 5,376 (71.5%) | 54.5 (36.1) | *p* < 0.001 |
| | Other | 641 (2.0%) | 31,662 (98.0%) | 16.1 (26.6) | *p* < 0.001 | 3,704 (30.8%) | 8,320 (69.2%) | 51.3 (34.7) | *p* < 0.001 |

Dx, diagnosis; NHB, non-Hispanic Black; NHW, non-Hispanic White; SD, standard deviation.

**Table 3. Unadjusted odds of undergoing surgical resection or receiving adjuvant chemotherapy.**

| Surgical resection, stage I to III (N = 353,998) | | | |
|---|---|---|---|
| **Factor** | **OR** | **95% CI** | **p-Value** |
| Insurance status | | | |
| Private | Ref | — | — |
| Medicare | 0.42 | 0.51 to 0.57 | <0.001 |
| Other government | 0.54 | 0.34 to 0.51 | <0.001 |
| Medicaid | 0.51 | 0.46 to 0.56 | <0.001 |
| Uninsured | 0.55 | 0.49 to 0.63 | <0.001 |
| Race/ethnicity | | | |
| NHW | Ref | — | — |
| NHB | 0.69 | 0.65 to 0.73 | <0.001 |
| Hispanic | 0.78 | 0.71 to 0.85 | <0.001 |
| Other | 0.91 | 0.82 to 1.02 | 0.10 |
| Age | 0.96 | 0.96 to 0.96 | <0.001 |
| Income | | | |
| Less than US$40,227 | Ref | — | — |
| US$40,227 to US$50,353 | 1.14 | 1.07 to 1.22 | <0.001 |
| US$50,353 to US$63,332 | 1.21 | 1.13 to 1.28 | <0.001 |
| US$63,333+ | 1.28 | 1.21 to 1.36 | <0.001 |
| Sex | | | |
| Male | Ref | — | — |
| Female | 0.95 | 0.91 to 0.99 | 0.18 |
| Rurality | | | |
| Metro | Ref | — | — |
| Urban | 1.04 | 0.97 to 1.10 | 0.26 |
| Rural | 1.40 | 1.16 to 1.67 | <0.001 |
| Charlson/Deyo comorbidity index | | | |
| 0 | Ref | — | — |
| 1 | 1.07 | 1.02 to 1.13 | 0.01 |
| 2 | 0.78 | 0.73 to 0.85 | <0.001 |
| 3 or more | 0.58 | 0.53 to 0.64 | <0.001 |
| Stage | | | |
| 1 | Ref | — | — |
| 2 | 2.81 | 2.67 to 2.95 | <0.001 |
| 3 | 3.83 | 3.62 to 4.05 | <0.001 |
| **Adjuvant chemotherapy, stage III (N = 129,341)** | | | |
| Insurance status | | | |
| Private | Ref | — | — |
| Medicare | 0.24 | 0.24 to 0.25 | <0.001 |
| Other government | 0.56 | 0.50 to 0.64 | <0.001 |
| Medicaid | 0.62 | 0.59 to 0.66 | <0.001 |
| Uninsured | 0.71 | 0.66 to 0.76 | <0.001 |
| Race/ethnicity | | | |
| NHW | Ref | — | — |
| NHB | 1.20 | 1.16 to 1.24 | <0.001 |
| Hispanic | 1.42 | 1.35 to 1.50 | <0.001 |
| Other | 1.38 | 1.31 to 1.46 | <0.001 |
| Age | 0.92 | 0.92 to 0.92 | <0.001 |

(*Continued*)

**Table 3.** (Continued)

| Surgical resection, stage I to III (*N* = 353,998) | | | |
|---|---|---|---|
| **Factor** | **OR** | **95% CI** | ***p*-Value** |
| Income | | | |
| Less than US$40,227 | Ref | — | — |
| US$40,227 to US$50,353 | 1.06 | 1.03 to 1.10 | <0.001 |
| US$50,353 to US$63,332 | 1.08 | 1.03 to 1.10 | <0.001 |
| US$63,333+ | 1.13 | 1.09 to 1.16 | <0.001 |
| Sex | | | |
| Male | Ref | — | — |
| Female | 0.83 | 0.0.81 to 0.85 | <0.001 |
| Rurality | | | |
| Metro | Ref | — | — |
| Urban | 1.06 | 1.03 to 1.10 | <0.001 |
| Rural | 1.04 | 0.96 to 1.12 | 0.35 |
| Charlson/Deyo comorbidity index | | | |
| 0 | Ref | — | — |
| 1 | 0.68 | 0.67 to 0.70 | <0.001 |
| 2 | 0.46 | 0.44 to 0.48 | <0.001 |
| 3 or more | 0.33 | 0.31 to 0.35 | <0.001 |
| Margin positive | | | |
| Negative | Ref | — | — |
| Positive | 0.77 | 0.74 to 0.80 | <0.001 |
| Number of lymph nodes resected | | | |
| ≤12 | Ref | — | — |
| ≥12 | 1.52 | 1.48 to 1.56 | <0.001 |

CI, confidence interval; NHB, non-Hispanic Black; NHW, non-Hispanic White; OR, odds ratio; Ref, reference.

In regard to receipt of adjuvant therapy in resected patients, NHB patients had a significantly decreased likelihood of receiving adjuvant chemotherapy [OR 0.83, 95% CI 0.78 to 0.87], but Hispanic patients actually had a higher likelihood of receiving adjuvant therapy [OR 1.20, 95% CI 1.09 to 1.33]. Compared to patients with private insurance, patients with Medicaid or no insurance also had a significantly decreased likelihood of receiving adjuvant chemotherapy compared to those with private insurance [OR 0.55, 95% CI 0.50 to 0.61(Medicaid), OR 46, 95% CI 0.41 to 0.53 (no insurance)], but those with Medicare did not (OR 1.02, 95% CI 0.98 to 1.08). (**Table 4**).

## Multivariable logistic regression: Joint effects

NHB and Hispanic patients with Medicare insurance had lower odds of receiving surgery compared to NHW patients with Medicare insurance [OR 0.59, 95% CI 0.53 to 0.66 (NHB); OR 0.71, 95% CI 0.61 to 0.84 (Hispanic)] (**Table 5**). Similar findings were also observed among NHB and Hispanic patients with private insurance compared to NHW patients with private insurance [OR 0.76, 95% CI 0.63 to 0.91 (NHB); OR 0.72, 95% CI 0.56 to 0.92 (Hispanic)]. The odds of receiving adjuvant chemotherapy was also lower for NHB compared to NHW among patients with Medicaid (OR 0.81, 95% CI 0.66 to 0.98), Medicare (OR 0.86, 95% CI 0.80 to 0.91), private insurance (OR 0.77, 95% CI 0.68 to 0.87), and other government insurance (OR 0.59, 95% CI 0.35 to 1.00). (**Table 6**) Hispanic patients actually had a higher

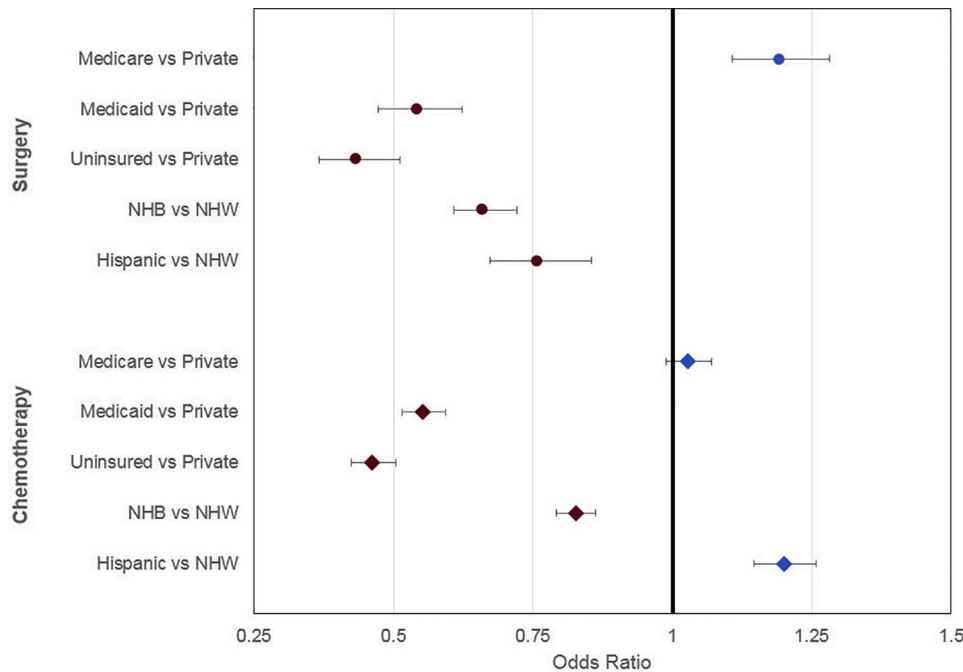

**Fig 2. Adjusted odds of receiving surgery or chemotherapy by insurance and race/ethnicity.** Data points represent OR, and bars represent 95% CI. Regression model also included the following covariates: age, sex, income, Charlson/Deyo comorbidity index, stage, grade, and rurality. For the chemotherapy group, margin status and number of nodes resected were also included. CI, confidence interval; NHB, non-Hispanic Black; NHW, non-Hispanic White; OR, odds ratio.

odds of receiving adjuvant chemotherapy compared to NHW patients in both the Medicare (OR 1.33, 95% CI 1.17 to 1.52) and Medicaid (OR 1.38, 95% CI 1.02 to 1.87) cohorts and were similar to NHW in the other insurance groups (**Table 6**).

## Discussion

Despite recent advancements in CC screening, diagnosis, and treatment, patients of NHB and Hispanic race/ethnicity continue to experience worse long-term outcomes. In this large, national study of over 300,000 patients with stage I, II, or III CC diagnosed at CoC hospitals, NHB and Hispanic patients had lower odds of undergoing curative-intent resection, and NHB had lower odds of receiving adjuvant chemotherapy, even in the setting of equivalent health insurance. Importantly, NHB patients had higher rates of no insurance or Medicaid insurance, lower median household income, and more often resided in a ZIP code with less educational attainment. Even after adjusting for these socioeconomic differences, NHB had lower odds of undergoing resection or receiving adjuvant chemotherapy. Further, these differences persisted when comparing racial cohorts with the same health insurance status, suggesting that adequate insurance coverage is not associated with mitigated racial disparities in cancer care delivery.

Across all stages of diagnosis, Black patients are less likely to receive treatment for colorectal cancer [15]. Prior studies of the Surveillance Epidemiology and End Results (SEER) registry have demonstrated that Black patients have lower odds of undergoing surgery for colorectal cancer [15–17]. Disparities in receipt of adjuvant chemotherapy for colorectal cancer and receipt of radiation for rectal cancer for Black and Hispanic patients have also been described based on SEER data [7,15–17]. A recently published study of California state registry data from 2000 to 2012 found that Black patients with metastatic colorectal cancer were less likely

**Table 4. Adjusted odds of undergoing surgical resection or receiving adjuvant chemotherapy.**

| Surgical resection, stage I to III ($N$ = 353,998) | | | |
|---|---|---|---|
| **Factor** | **OR** | **95% CI** | **$p$-Value** |
| Insurance status | | | |
| Private | Ref | — | — |
| Medicare | 1.19 | 1.11 to 1.28 | <0.001 |
| Other government | 0.55 | 0.44 to 0.70 | <0.001 |
| Medicaid | 0.54 | 0.47 to 0.62 | <0.001 |
| Uninsured | 0.43 | 0.37 to 0.51 | <0.001 |
| Race/ethnicity | | | |
| NHW | Ref | — | — |
| NHB | 0.66 | 0.61 to 0.72 | <0.001 |
| Hispanic | 0.76 | 0.67 to 0.85 | <0.001 |
| Other | 0.87 | 0.79 to 0.97 | 0.002 |
| Age | 0.94 | 0.94 to 0.95 | <0.001 |
| Income | | | |
| Less than US$40,227 | Ref | — | — |
| US$40,227 to US$50,353 | 1.05 | 0.96 to 1.14 | 0.29 |
| US$50,353 to US$63,332 | 1.09 | 1.00 to 1.19 | 0.059 |
| US$63,333+ | 1.22 | 1.11 to 1.34 | <0.001 |
| Sex | | | |
| Male | Ref | — | — |
| Female | 1.04 | 0.98 to 1.09 | 0.18 |
| Rurality | | | |
| Metro | Ref | — | — |
| Urban | 1.01 | 0.93 to 1.10 | 0.79 |
| Rural | 1.53 | 1.23 to 1.90 | <0.001 |
| Charlson/Deyo comorbidity index | | | |
| 0 | Ref | — | — |
| 1 | 1.19 | 1.11 to 1.26 | <0.001 |
| 2 | 0.95 | 0.87 to 1.04 | 0.27 |
| 3 or more | 0.73 | 0.65 to 0.81 | <0.001 |
| Stage | | | |
| 1 | Ref | — | — |
| 2 | 2.91 | 2.69 to 3.15 | <0.001 |
| 3 | 3.88 | 3.58 to 4.21 | <0.001 |
| **Adjuvant chemotherapy, stage III ($N$ = 129,341)** | | | |
| Insurance status | | | |
| Private | Ref | — | — |
| Medicare | 1.02 | 0.98 to 1.08 | 0.26 |
| Other government | 0.83 | 0.67 to 1.03 | 0.084 |
| Medicaid | 0.55 | 0.50 to 0.61 | <0.001 |
| Uninsured | 0.46 | 0.41 to 0.53 | <0.001 |
| Race/ethnicity | | | |
| NHW | Ref | — | — |
| NHB | 0.83 | 0.78 to 0.87 | <0.001 |
| Hispanic | 1.20 | 1.09 to 1.33 | <0.001 |
| Other | 0.97 | 0.91 to 1.04 | 0.43 |
| Age | 0.90 | 0.90 to 0.91 | <0.001 |

(*Continued*)

**Table 4.** (Continued)

| Surgical resection, stage I to III ($N$ = 353,998) | | | |
|---|---|---|---|
| **Factor** | **OR** | **95% CI** | **_p_-Value** |
| Income | | | |
| Less than US$40,227 | Ref | — | — |
| US$40,227 to US$50,353 | 1.07 | 1.01 to 1.12 | 0.015 |
| US$50,353 to US$63,332 | 1.17 | 1.11 to 1.23 | <0.001 |
| US$63,333+ | 1.22 | 1.16 to 1.29 | <0.001 |
| Sex | | | |
| Male | Ref | — | — |
| Female | 1.01 | 0.98 to 1.04 | 0.39 |
| Rurality | | | |
| Metro | Ref | — | — |
| Urban | 0.97 | 0.92 to 1.03 | 0.35 |
| Rural | 0.94 | 0.84 to 1.04 | 0.24 |
| Charlson/Deyo comorbidity index | | | |
| 0 | Ref | — | — |
| 1 | 0.85 | 0.82 to 0.88 | <0.001 |
| 2 | 0.64 | 0.60 to 0.67 | <0.001 |
| 3 or more | 0.47 | 0.44 to 0.51 | <0.001 |
| Margin positive | | | |
| Negative | Ref | — | — |
| Positive | 0.77 | 0.72 to 0.81 | <0.001 |
| Number of lymph nodes resected | | | |
| $\leq$12 | Ref | — | — |
| $\geq$12 | 1.28 | 1.22 to 1.34 | <0.001 |

CI, confidence interval; NHB, non-Hispanic Black; NHW, non-Hispanic White; OR, odds ratio; Ref, reference.

to receive chemotherapy or to undergo hepatic metastectomy [18]. A recent study of patients with gastrointestinal cancers (including colorectal cancer) identified in the 2004 to 2015 NCDB found that a disparity in the receipt of surgery had significant influence on survival disparity for Black compared to White patients [9]. In addition, Black patients are less likely to enroll in clinical trials and are less likely to discuss or consider trial enrollment [19,20]. Black patients are also less likely to receive posttreatment surveillance testing [21]. The aggregate disparity in receipt of care for Black patients appears to correlate with the ultimate disparity in survival outcomes for these same patients [10,21,22].

It is also well established that minority race/ethnicity patients are more frequently underinsured. Nationally, Black and Hispanic patients have lower rates of private insurance and concurrently higher rates of public or no insurance compared to White patients [23]. Uninsured rates are particularly high among rural residents of racial/ethnic minority and correlate with self-reported poor health [24]. Inadequate insurance not only limits receipt of care but may also even impact the potential therapeutic benefit of experimental therapy in the context of clinical trials. Pooled data from clinical trials found that patients with Medicaid insurance or with no insurance received less benefit from experimental therapy in the context of a clinical trial when compared to patients with Medicare or private insurance [25]. Not surprisingly, those with Medicaid or no insurance included higher percentages of minority race/ethnicity.

**Table 5. Effect modification of insurance on race/ethnicity and surgical resection.**

| Insurance | Race/ethnicity | Surgical resection | | | |
|---|---|---|---|---|---|
| | | OR | 95% CI | p-Value | E-Value |
| Uninsured | NHW | Ref | | | |
| | NHB | 0.91 | 0.64 to 1.28 | 0.58 | 1.28 |
| | Hispanic | 0.95 | 0.6 to 1.50 | 0.81 | 1.2 |
| Medicaid | NHW | Ref | | | |
| | NHB | 0.94 | 0.73 to 1.20 | 0.60 | 1.22 |
| | Hispanic | 1.27 | 0.91 to 1.77 | 0.15 | 1.51 |
| Medicare | NHW | Ref | | | |
| | NHB | 0.59 | 0.53 to 0.66 | <0.001 | 1.92 |
| | Hispanic | 0.71 | 0.61 to 0.84 | <0.001 | 1.65 |
| Private | NHW | Ref | | | |
| | NHB | 0.76 | 0.63 to 0.91 | 0.004 | 1.55 |
| | Hispanic | 0.72 | 0.56 to 0.92 | 0.009 | 1.64 |
| Other government | NHW | Ref | | | |
| | NHB | 0.98 | 0.49 to 1.95 | 0.95 | 1.2 |
| | Hispanic | 0.44 | 0.16 to 1.21 | 0.11 | 2.39 |

CI, confidence interval; NHB, non-Hispanic Black; NHW, non-Hispanic White; OR, odds ratio; Ref, reference.

Insurance coverage disparities in general are associated with inadequate CC care and survival and represent a key contributing factor to outcome disparities for patients of minority race and ethnicity [10,26]. A 2016 study on the Massachusetts health insurance reform from 2006 identified improved colorectal cancer resection rates in the state compared to 3 control states without similar health insurance reforms [27]. Although an association with racial treatment disparities was not specifically examined, these findings, along with other studies

**Table 6. Effect modification of insurance on race/ethnicity and adjuvant chemotherapy.**

| Insurance | Race/ethnicity | Adjuvant chemotherapy | | | |
|---|---|---|---|---|---|
| | | OR | 95% CI | p-Value | E-value |
| Uninsured | NHW | Ref | | | |
| | NHB | 0.96 | 0.72 to 1.29 | 0.81 | 1.15 |
| | Hispanic | 1.07 | 0.76 to 1.50 | 0.70 | 1.22 |
| Medicaid | NHW | Ref | | | |
| | NHB | 0.81 | 0.66 to 0.98 | 0.031 | 1.47 |
| | Hispanic | 1.38 | 1.02 to 1.87 | 0.035 | 1.63 |
| Medicare | NHW | Ref | | | |
| | NHB | 0.86 | 0.80 to 0.91 | <0.001 | 1.39 |
| | Hispanic | 1.33 | 1.17 to 1.52 | <0.001 | 1.58 |
| Private | NHW | Ref | | | |
| | NHB | 0.77 | 0.68 to 0.87 | <0.001 | 1.54 |
| | Hispanic | 0.96 | 0.81 to 1.13 | 0.64 | 1.16 |
| Other government | NHW | Ref | | | |
| | NHB | 0.59 | 0.35 to 1.00 | 0.05 | 1.92 |
| | Hispanic | 0.96 | 0.81 to 1.13 | 0.64 | 1.16 |

CI, confidence interval; NHB, non-Hispanic Black; NHW, non-Hispanic White; OR, odds ratio; Ref, reference.

investigating the impact of the ACA, indicate that insurance coverage plays an important role in the observed treatment and survival disparities in colorectal cancer [28,29].

However, insurance is not the only factor. Evidence indicates that disparities in long-term outcomes experienced by Black patients are multilevel in etiology and may include limited access to screening, mistrust of physicians, socioeconomic barriers including financial limitations, and receipt of quality care [3,30]. This study sought specifically to investigate the intersection of race/ethnicity and insurance with cancer treatment disparities. To our knowledge, the only prior similar analysis on cancer treatment utilized the SEER dataset from 1990 to 2010 and found that Black patients had lower odds of receipt of adjuvant chemotherapy regardless of insurance status [7]. However, there are significant limitations in the assessment of chemotherapy use within the SEER dataset. A 2016 study on disparities in minimally invasive surgery (MIS) approach for colorectal surgery did find persistent Black disparities after stratification by private versus public insurance; however, indications for surgery included benign colorectal and diverticular disease. [31]. While other studies have attempted to adjust for either insurance or race/ethnicity as a covariate, this intersection of insurance and race/ethnicity on cancer treatment disparities has not been directly explored. In this analysis, NHB and Hispanic patients had a persistently lower odds of surgical resection. Interestingly, however, although NHB who underwent resection had lower odds of receiving adjuvant chemotherapy, Hispanic patients did not even in the setting of equivalent health insurance status. This was a surprising finding and suggests that independent factors may play a role in explaining disparities among different races as well as different treatment regimens even among underrepresented and underprivileged minorities.

The use of data obtained from the NCDB merits consideration of several limitations [32]. First, continuity or disruption of insurance coverage cannot be assessed within NCDB; therefore, the association between outcomes and interrupted coverage or disruption of preexisting coverage remains unknown. Second, specific details on chemotherapy agents or dosing are not available to assess for standard of care treatment. Third, although the NCDB is based on rigorous comprehensive data collection, the dataset lacks information regarding specific SDOH, thereby limiting a more comprehensive analysis of other social factors likely to affect healthcare access. In addition, the available socioeconomic variables are based on median values from the ZIP code of residence and are not specific to the individual patient. Furthermore, many potentially confounding factors that may help explain findings in this study are not collected within the NCDB. Fifth, reasons for why a specific treatment was not readily available within the dataset. Finally, the NCBD is not inclusive of all cancer care facilities, hence the data presented may not be generalizable to non CoC-accredited facilities.

## Conclusions

Patients with Medicaid insurance coverage or lack of insurance and patients of minority race/ethnicity, especially NHB, are less likely to undergo surgical resection or receive adjuvant chemotherapy. Black and Hispanic patients with equivalent insurance coverage still experience lower odds of surgical resection, and Black patients still experience lower odds of receipt of adjuvant chemotherapy. Changes in health policy must recognize that provision of insurance alone is not associated with improved disparities in cancer care among minority populations and that different minority populations may have different challenges precluding receipt of the standard of care. Comprehensive study of other SDOH such as poverty, literacy, and rurality of residence, as well as policy change addressing these factors, is needed to ensure equity in cancer patient care for patients of all races.

## Supporting information

**S1 Table. Prospective analysis plan.**
(DOCX)

**S2 Table. STROBE Checklist. STROBE, Strengthening the Reporting of Observational Studies in Epidemiology.**
(DOCX)

## Acknowledgments

We would like to acknowledge Josh Yang and Reba Bullard for their assistance in the preparation of this manuscript.

## Disclaimers

The National Cancer Database (NCDB) and the hospitals participating in the NCDB are the source of the data used herein. The NCDB is a joint project of the Commission on Cancer (CoC) of the American College of Surgeons and the American Cancer Society. The data used in the study are derived from a deidentified NCDB file. The American College of Surgeons and the CoC have not verified and are not responsible for the analytic or statistical methodology employed or the conclusions drawn from these data by the investigator.

## Meeting presentation

Presented in oral format at the virtual American College of Surgeons 2020 Clinical Congress, October 2020.

## Author Contributions

**Conceptualization:** Scarlett Hao, Rebecca A. Snyder, Alexander A. Parikh.

**Data curation:** Scarlett Hao, William Irish, Alexander A. Parikh.

**Formal analysis:** William Irish.

**Investigation:** Scarlett Hao, Rebecca A. Snyder, Alexander A. Parikh.

**Methodology:** Scarlett Hao, Rebecca A. Snyder, William Irish, Alexander A. Parikh.

**Project administration:** Rebecca A. Snyder, Alexander A. Parikh.

**Resources:** Scarlett Hao, Rebecca A. Snyder, Alexander A. Parikh.

**Software:** William Irish.

**Supervision:** Rebecca A. Snyder, Alexander A. Parikh.

**Validation:** Rebecca A. Snyder.

**Writing – original draft:** Scarlett Hao.

**Writing – review & editing:** Rebecca A. Snyder, William Irish, Alexander A. Parikh.

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
