## [Editor Report · Decision Letter 0]

10 May 2021

Dear Dr Parikh, 

Thank you for submitting your manuscript entitled "Intersection of minority race and health insurance in treatment disparities of colon cancer" for consideration by PLOS Medicine.

Your manuscript has now been evaluated by the PLOS Medicine editorial staff and I am writing to let you know that we would like to send your submission out for external peer review.

Kind regards,

Beryne Odeny

Associate Editor

PLOS Medicine

---

## [Decision Letter · Decision Letter 1]

22 Jun 2021

Dear Dr. Parikh,

Thank you very much for submitting your manuscript "Intersection of minority race and health insurance in treatment disparities of colon cancer" (PMEDICINE-D-21-02061R1) for consideration at PLOS Medicine. 

[LINK]

Considering these reviews, we would be grateful if you could please revise your manuscript to respond to comments raised by reviewers. We would strongly recommend that you pay special attention to cross-cutting reviewer concerns such as the inability to account for individual level variables such as education. Please note that this is not a guarantee that we will accept the manuscript and that further consideration is dependent on the submission of a manuscript that addresses all reviewer concerns. We will carefully review your manuscript upon revision, so please ensure that your revision is thorough.

We expect to receive your revised manuscript by Jul 13 2021 11:59PM. Please email us (plosmedicine@plos.org) if you have any questions or concerns.

We look forward to receiving your revised manuscript. 

Sincerely,

Beryne Odeny, 

PLOS Medicine 

plosmedicine.org

Editorial comments:

1) Please revise your title according to PLOS Medicine's style. Your title must be nondeclarative and not a question. It should begin with main concept if possible. "Effect of" or “risk of” should be used only if causality can be inferred, i.e., for an RCT. Please include the setting. In addition, please place the study design ("A retrospective study,") in the subtitle (i.e., after a colon).

2) Abstract:

a) Please ensure that all numbers presented in the abstract are present and identical to numbers presented in the main manuscript text.

b) Please include the study setting.

c) Please include the specific important dependent variables that are adjusted for in the analyses.

d) Please quantify the main results (with 95% CIs and p values).

3) Did your study have a prospective protocol or analysis plan? Please state this (either way) early in the Methods section. 

4) For this observational study, in the manuscript text, please indicate: (1) the analytical methods by which you planned to test your hypothesis, (2) the analyses you actually performed, and (3) when reported analyses differ from those that were planned, transparent explanations for differences that affect the reliability of the study's results. If a reported analysis was performed based on an interesting but unanticipated pattern in the data, please be clear that the analysis was data-driven.

5) In statistical methods, please refer to any post-hoc corrections to correct for multiple comparisons during your statistical analyses. If these were not performed please justify the reasons. Please refer to our statistical reporting guidelines for assistance (https://journals.plos.org/plosone/s/submission-guidelines.#loc-statistical-reporting)

6) Thank you for providing your STROBE checklist. Please replace the page numbers with paragraph numbers per section (e.g. "Methods, paragraph 1"), since the page numbers of the final published paper may be different from the page numbers in the current manuscript.

7) Your study is observational and therefore causality cannot be inferred. Please remove language that implies causality. Instead, refer to associations consistently throughout the text.

8) Some reviewer concerns include the lack of individual level SES and some health care variables. These might affect inference and policy recommendations with regards to health system role vs other drivers of disparity. Please consider some sensitivity analyses to address this or at minimum calculate the e-value to determine how large the confounders would have to be to invalidate the race effect.

9) In your statistical analyses, please use hierarchical/ multilevel models or generalized estimating equations given that nationwide data is likely clustered at state/ county and hospital levels. The potential clustering of data (e.g., among patients from the same locality or hospital) would result in spurious effect estimates and standard errors.

10) Please provide 95% CIs and p values for all estimates in the text and tables.

11) Please specify the significance level used (eg, P<0.05, two-sided) and the statistical test used to derive a p value.

12) Please do not report P<0.01; report as P < 0.001.

13) Please define the abbreviations in Tables and Figure e.g., AJCC, NOS, HS,Dx, SD, etc.

14) Please indicate in the figure caption the meaning of the bars and whiskers in Figure 2

Comments from the reviewers:

Reviewer #1: This study aims to determine the interaction of race and insurance with CC treatment disparities. 

Comments:

The authors have appropriately provided the STROBE checklist in the supplementary material.

"The National Cancer Database (NCDB) sponsored by the American College of Surgeons and American Cancer Society gathers data from more than 1500 Commission on Cancer (CoC)-accredited facilities in the United States."

Did the authors consider accounting for clustering by health care facility within the analysis?

"The primary outcomes of interest were: 1) receipt of surgical resection and 2) receipt 124 of adjuvant chemotherapy in the subgroup of eligible patients with resected stage III CC, 125 stratified by race and insurance."

Can the authors please comment on whether it is possible to attain information on the offer of treatment to patients, and to therefore model patient uptake rates accounting for this?

"Continuous variables are described by the number of non-missing observations, mean, standard deviation, median, and 25th and 75th percentiles. Categorical variables are described overall and by cohort with the number of patients and percentage for each category. Missing data was considered as a separate category".

The authors have used valid statistical descriptors for the data types in hand.

"To adjust for confounding and estimate the association of outcome to covariates, data was fit using multivariable binary logistic regression models. Two models were fit to the data: a main effects model with additive terms for race and insurance status adjusted for additional covariates and a joint effects model with a two way interaction term for race and insurance also adjusted for additional covariates. These included: age, race, sex, insurance status, income level, education, rurality, comorbidity, distance traveled for care, and tumor grade. The joint effects model was used to evaluate the effect of race on outcome within levels of insurance status. Adjusted odds ratios (OR) and 95% confidence intervals (CI) are provided as measures of strength of association and precision, respectively. The joint effect of race and insurance status on outcomes was tested using Wald's Chi-square on 12 degrees of freedom."

With reference to my earlier comment regarding possible clustering in the data by health care facility, did the authors consider a modelling approach (such as multilevel modelling, for example) that takes this into account? 

The authors have done well to include covariates in an attempt to adjust for potential confounding in the models. Can they further comment on whether a measure of deprivation is available for inclusion?

Tables 1 and 2: Did the authors consider statistically testing for differences between racial cohorts and insurance groups, and presenting these results within the Tables and in support of statements in the Results text?

Can the authors please comment on and discuss how "When evaluated by race, 67.6% of patients of NHW race received adjuvant chemotherapy compared to 70.9% of patients of NHB race" (Table 2) matches up with the reduced odds seen in Tables 3 and 4?

Reviewer #2: Thank you for the opportunity to review this paper about the intersection of race and health insurance in the treatment of colon cancer. This paper makes an important contribution to the literature by examining whether racial disparities persist after stratifying by health insurance coverage. Below are suggestions for improving the paper. 

* Throughout the paper, the authors use the term "non-White." This term centers whiteness and likely categorizes patients in a way they would not self-identify. Furthermore, using this term may imply to some readers that patients of all races other than white were grouped together in analyses, which does not appear to be the case for this study. Please consider being more precise about the racial and ethnic groups of patients (e.g., non-Hispanic Black and Hispanic) when describing study results. 

* In the methods, please clarify the approach used to address missing data. 

* In the limitations, it is also important to note that some variables (e.g., educational attainment) were derived using data from the patient's zip code and may not accurately represent the patient's actual education level. 

* In the discussion, the authors note that, "Third, although the NCDB is based on rigorous comprehensive data collection, the dataset lacks information regarding specific social determinants of health, thereby limiting a more comprehensive analysis of other social factors likely to affect health care access." This is an important point that warrants more discussion in the paper (e.g., in the introduction and conclusion). It may be helpful to provide key examples of prior evidence about social factors and determinants of health (e.g., racism in health care) that likely influence these outcomes. It is important to be specific about these factors so that readers will not make inaccurate assumptions about potential causes of racial disparities in cancer outcomes. 

* In the conclusion, please clarify what is meant by "inadequate" insurance coverage. 

* Please revise the following sentence as there appears to be a typo: Uninsured rates are particularly high among rural residents of non-White racial minority and correlate with high rates of self-reported poor health.

Reviewer #3: Dear authors,

This is a well-written manuscript assessing the associations between race/ethnicity, insurance status and colon cancer outcomes. The study's main findings are than non-Hispanic Black patients are less likely to undergo resection for colon cancer or to receive chemotherapy in comparison to non-Hispanic white patients. The study's strongest strength is the use of a large, nationally-representative dataset. Additionally, the manuscript is well organized and written clearly enough to be accessible to health professionals that do not specialize in colon cancer research. However, the study has a number of weaknesses, including the lack of data on important confounders, such as individual-level SES and information about the patients' health systems. Also, the study contained limited granularity in the number of racial and ethnic groups examined. 

Importantly, this manuscript is unlikely to directly or substantially affect public health policy. The racial disparity experienced by non-Hispanic Blacks in health care access and colorectal cancer screening, diagnosis and treatment has been well-documented in the literature. The results of this study do not provide a substantial advance over existing knowledge in colorectal cancer outcomes research.

Specific suggestions for the manuscript are provided below:

1. Abstract - the second sentence refers to "these disparities", however disparities have not yet been defined in the text. Please rephrase to define the disparities of interest for the manuscript.

2. General - please be consistent in the use of colon vs colorectal cancer

3. General - please use ethnicity (not race) when referring to comparisons to Hispanic patients 

4. Introduction - please update 1st paragraph with data from 2021, and there are racial groups that have lower CC rates than non-Hispanic whites (e.g. API), please revise. 

5. Introduction - 2nd paragraph: it is not useful to describe all non-white Americans as a monolith. Please be more specific about the disparities being addressed.

6. Methods - does the dataset include indication for why patients did not receive surgery/chemotherapy?

7. Methods/Discussion - educational attainment measured at the zip code-level was included as a covariate. Please discuss how this variable may be associated with the outcomes of the study.

[LINK]

---

## [Decision Letter · Decision Letter 2]

16 Sep 2021

Dear Dr. Parikh,

Thank you very much for re-submitting your manuscript "Association of race and health insurance in treatment disparities of colon cancer: A retrospective analysis utilizing a population database." (PMEDICINE-D-21-02061R2) for review by PLOS Medicine.

I have discussed the paper with my colleagues and the academic editor and it was also seen again by two reviewers. I am pleased to say that provided the remaining editorial and production issues are dealt with we are planning to accept the paper for publication in the journal.

[LINK]

We look forward to receiving the revised manuscript by Sep 23 2021 11:59PM.   

Sincerely,

Beryne Odeny, 

Associate Editor 

PLOS Medicine

plosmedicine.org

Requests from Editors:

1) Please include the study setting/ country in the title.

2) Please include the study country in the abstract’s “Method and Findings” section. 

3) Please integrate the author summary with the main text. This should follow the abstract.

4) In the main text and tables, please provide both Odds Ratios (OR) and adjusted ORs for unadjusted and adjusted analyses.

5) Line #161 “… some…” instead of “…som…”

6) References - Please ensure that journal name abbreviations consistently match those found in the National Center for Biotechnology Information (NCBI) databases, and are appropriately formatted and capitalized. https://journals.plos.org/plosmedicine/s/submission-guidelines#loc-references. 

Comments from Reviewers:

Reviewer #1: The authors have satisfactorily responded to each comment in turn, amending the analytical approach and presenting statistical tests accordingly.

Reviewer #2: Thank you for the opportunity to review the revised version of this manuscript. The authors have addressed my previous concerns and suggestions. This paper will make a useful contribution to the field, especially now that the authors revised their analytic approach and clarified a number of points throughout the paper.

[LINK]

---

## [Editor Report · Decision Letter 3]

8 Oct 2021

Dear Dr Parikh, 

On behalf of my colleagues and the Academic Editor, Dr. Margaret Kruk, I am pleased to inform you that we have agreed to publish your manuscript "Association of race and health insurance in treatment disparities of colon cancer: A retrospective analysis utilizing a national population database in the United States." (PMEDICINE-D-21-02061R3) in PLOS Medicine.

PRESS

Sincerely, 

Beryne Odeny 

PLOS Medicine